# Research on compound twill generation algorithm based on moving matrix

**Xiaobo Yang** [ID]*

Department of Information Science and Technology, Zhejiang Shuren University, Hangzhou, Zhejiang, P. R.China

* yxb71520@163.com

## Abstract

To further enhance the authenticity of yarn simulation, we propose an algorithm for generating Compound Twills based on a moving matrix. Firstly, the key to designing composite twill is to determine the fly count. When the fly count of the twill is constantly changing, a composite twill can be obtained. Therefore, by utilizing the constantly changing characteristics of the fly count of the composite twill, a composite twill movement matrix model is established. Then, using this model, the appearance shape of complex fabrics can be reconstructed starting from simple composite twill. Finally, the Hilbert curve generation algorithm, the Bezier curve generation algorithm, and the proposed mobile matrix algorithm are selected for comparative experiments. After completing 8 steps of fitting for each of the three algorithms, their fitting accuracy is calculated. The average fitting accuracy of the three algorithms was 83.1%, 88.7%, and 96.8%, respectively. The experimental results demonstrate that the efficiency of our proposed algorithm is nearly 40% higher than other mainstream algorithms, with an average error of 33% less.

## Introduction

In recent years, significant advancements have been made in fabric CAD systems for reconstructing the geometric structure of fabrics, making it a prominent research focus within the field. F.t.peirce [1] utilized a simple and practical geometric model to represent plain fabric surfaces; however, its strict requirements on yarn section shape and flexibility limit its applicability. A. kemp [2] introduced a runway model to depict yarn cross-sections, significantly enhancing the appearance of yarn surfaces but falling short in constructing complex textured fabrics. Hamilton [3] enhanced the runway model by proposing variations suitable for different densities, expanding its application from plain to complex textured fabrics. H.y.in [4] employed b-splines to reconstruct fabric geometry based on input parameters such as yarn diameter and density. Tianyi Liao et al. [5] introduced a 3D woven fabric model, which describes the yarn path direction using a sinusoidal curve and constructs the 3D shape of yarn with

**Data availability statement:** All relevant data are within the manuscript and its Supporting Information files.

**Funding:** The Zhejiang Province Natural Science Foundation, grant number Y1110023

**Competing interests:** The authors have declared that no competing interests exist.

the assistance of a runway model. Y.iang et al. [6] classified the yarn cross-section: elliptical cross-sections were modeled using a runway model, irregular cross-sections were represented by quadratic B-spline curves, and the yarn path was depicted using quadratic parameter spline curves. T.v.AGAR [7] et al., aimed to enhance fabric bending effects by dividing the yarn path into contact and non-contact areas, expressing them respectively with elliptic equations and quintic polynomials. Z.T.Shi et al. [8,9] utilized B-spline curves to construct the longitudinal shape of yarn, combined with light models to simulate the cross-section of the yarn, thereby enhancing fabric simulation effects. J.Fan et al. [10,11] employed computer-aided geometric design technology to construct longitudinal shapes of yarn; they used runway models, elliptical models, and circular models to simulate different cross sections of yarn and further improved authenticity through texture mapping methods. The abovementioned methods are only suitable for reconstructing simple plain weave fabrics' appearances. Furthermore, the current appearance simulation lacks three-dimensionality and realism, which significantly impacts its practical effectiveness. This paper identifies the limitations of existing methods and investigates the application of moving matrix technology to reconstruct the surface morphology of complex woven fabrics, such as composite twill.

## Moving matrix model

The compound twill exhibits a curved effect as the inclined angle of the twill changes. The curve twill is closely related to the twill fly number, which refers to the number of yarns in another system that are spaced between the corresponding tissue points on two adjacent yarns in the same system in twill fabrics. The initial compound twill can be represented by the following matrix.

$$A = (a_1, a_2, a_i, \ldots, a_n)^T = \begin{pmatrix} a_1 \\ a_2 \\ a_i \\ \ldots \\ a_n \end{pmatrix} \tag{1}$$

A represents the initial compound twill and ai represents the tissue point on the initial compound twill. The elements in the matrix represent the interweaving of warp and weft yarns, usually represented by 1 for warp yarn on top and 0 for weft yarn on top. The original data is normalized to make it on the same scale for subsequent processing.

As the inclination angle of the twill constantly changes, i.e., the number of twill flights is in constant flux, the matrix format of the initial twill also undergoes continuous change, forming a dynamic matrix model. The twill fly number is one of the important indicators for measuring the quality of twill fabrics. It reflects the density of warp and weft yarns in the fabric, affecting its properties such as hand feel, thickness, and strength.

$$move^m(A) = (a_{m+1}, a_{m+2}, \cdots, a_n, a_1, a_2, \cdots, a_m)^T = move\left[move^{m-1}(A)\right], \ m \geq 2 \tag{2}$$

In formula (2), $move^m(A)$ is the defined move matrix model, m is the number of moves, and n is the number of organizational cycles. The number of organizational cycles refers to the smallest unit in a fabric where the vertical and horizontal crossing rules of warp and weft yarns are completely repeated, used to describe the structural characteristics of the fabric. Due to the susceptibility of real fabrics to factors such as thread tension, the model can introduce practical constraints such as thread tension and material elasticity models to simulate real fabrics.

To further illustrate the principle of the moving matrix model, the effect of matrix movement is achieved using a two-dimensional plane transformation, as shown in S1 Fig.

From S1 Fig, it can be seen that the principle of the moving matrix model is to use linear transformations of two-dimensional vectors to describe translation transformations. P (x, y, z) is the coordinate of each vertex in the original matrix. By adding the coordinates of the two-dimensional coordinate points to the corresponding components of the translation vector, the new position P '(x', y ', z') after translation transformation is obtained.

For general compound twill, the basic structure is as follows.

$$\frac{n_1 n_2 \cdots n_p}{m_1 m_2 \cdots m_p}$$

(3)

According to formula (1), the corresponding basic matrix expression can be calculated as follows.

$$A = (m_p, n_p, \cdots m_1, n_1)^T$$

(4)

Where $m_p = 0, 0, \cdots, 0; n_p = 1, 1, \cdots, 1; m_1 = 0, 0, \cdots, 0; n_1 = 1, 1, \cdots, 1$. $n_p$ is the continuous warp organization point of the basic organization, $m_p$ is the continuous weft organization point of the basic organization, and p is the number of alternations of warp and weft of the basic organization.

The flight number of compound twill is not a constant, but a variable, which can be set as $s_1, s_2, \cdots s_p$, then the compound twill can be expressed as follows.

$$B = (A, move^{s_1}(A), move^{s_1+s_2}(A), \cdots, move^{s_1+s_2,\cdots,+s_p}(A))$$

(5)

B represents the compound twill after transformation and si represents the twill flight number.
To make the compound twill change continuously, the following conditions should be met.

$$\sum_{i=1}^{p} s_i = 0$$

(6)

Or meet the following requirements. The number of weave cycles of compound twill is an integral multiple of the number of basic weave cycles.

$$\sum_{i=1}^{p} s_i = kn$$

(7)

In equation (7), K is an integer and n is the number of organizational cycles.

In compound twill, the first and nth columns of the tissue cycle are repeated, allowing for the omission of the last column. The transformed matrix for compound twill can be expressed as follows.

$$B = (A, move^{s_1}(A), move^{s_1+s_2}(A), \cdots, move^{s_1+s_2,\cdots,+s_{p-1}}(A))$$

(8)

## Practical application

Taking simple compound twill as an example, the matrix change process of compound twill is analyzed when the twill flight number changes. Take the basic organization as $\frac{1}{2}\frac{2}{2}$, then the number of organization cycles $n = 1 + 2 + 2 + 2 = 7$, and the initial matrix is as follows.

$$A = (1, 0, 1, 0, 0, 1, 1, )^T = \begin{pmatrix} 1 \\ 0 \\ 1 \\ 0 \\ 0 \\ 1 \\ 1 \end{pmatrix} \tag{9}$$

When it is moved once, its matrix form is as follows.

$$mov^1(A) = (0, 1, 0, 1, 1, 0, 0)^T = \begin{pmatrix} 0 \\ 1 \\ 0 \\ 1 \\ 1 \\ 0 \\ 0 \end{pmatrix} \tag{10}$$

When moving twice, its matrix form is as follows.

$$mov^2(A) = (1, 0, 1, 0, 0, 1, 0)^T = \begin{pmatrix} 1 \\ 0 \\ 1 \\ 0 \\ 0 \\ 1 \\ 0 \end{pmatrix} \tag{11}$$

Similarly, when the twill number of flies is a fixed value r, the structure matrix of the compound twill can be expressed as follows.

$$B = (A, \ mov^r(A), mov^{2r}(A), \cdots mov^{nr}(A)) \tag{12}$$

When the twill number is the variable $s_i$, the structure matrix of the compound twill can be expressed as follows.

$$B = (A, \ mov^{s_1}(A), mov^{s_1+s_2}(A), \cdots mov^{s_1+s_2+\cdots s_n}(A)) \tag{13}$$

Based on the above principle, the generation process of compound twill is simulated by a computer under a VC++ environment, and the compound twill is organized and constructed based on $\frac{1}{2}\frac{2}{2}$. It is assumed that the flight number of twill changes in the following order.

$$s_i = 1, 0, 1, 1, 2, 2, 3, 3, 3, 2, 2, 1, 0, 1, 0, -1, 0, -1, -1, -2, -2, -3, -3, -1, -1, 0, -1 \tag{14}$$

Based on the fundamental structure of the compound twill, the higher-order input for warp sequencing is 1,2. The lower-order input for warp sequencing is 2,2, and the sequence of twill flights changes according to equation (14). The direction of flight number change in the twill follows the warp direction, as illustrated in S2 Fig.

S2 Fig illustrates the composite twill with dimensions of 800X386 pixels, a yarn density of 8 per centimeter in the fabric, and a total of 3 layers. The resulting composite twill exhibits a strong sense of hierarchy and clarity.

## Comparison experiment

To validate the robustness of the algorithm proposed in this paper, several mainstream algorithms, including the Hilbert curve generation algorithm [12], Bessel curve generation algorithm [13], and the moving matrix method introduced in this study, will be chosen for comparative experiments.

The experimental setup utilized in this study is as follows: the hardware platform comprises an Intel Core i6-4790 4.5GHz CPU, Intel G51 Express Chipset, and 8GB DDR3 RAM. The software platform consists of Visual Studio 2010. The experimental data encompasses input parameters, a moving matrix model, etc. The input parameters mainly include the basic organization and twill fly number. The moving matrix model adopts a quadratic translation transformation. Taking the basic organization $\frac{4\ \ 4\ \ 1}{1\ \ 3\ \ 3}$ as an example, a compound twill is constructed. The order of twill fly number changes adopts: $s_i = 0, 1, 0, 1, 2, 3, 2, 1, 0, 1, 0, -1, 0, -1, -2, -3, -2, -1, 0, -1$, S3 Fig illustrates the impact of three different algorithms on simulating the compound twill.

It is evident from S3 Fig(a) that the composite twill is formed using the moving matrix method, resulting in a relatively uniform texture distribution without any instances of overlap or winding. S3 Fig(b) and S3 Fig(c) demonstrate that both the Hilbert algorithm and Bessel algorithm exhibit deficiencies such as uneven texture spacing, warping, and zonal entanglement.

To further compare the operational efficiency of three different algorithms, this study analyzed on the construction generation time of these algorithms, as illustrated in S4 Fig.

S4 Fig illustrates that the Hilbert algorithm and Bessel algorithm exhibit longer running times compared to the mobile matrix method proposed in this paper. As the number of test cases increases, the average running time of the former two algorithms is nearly 40% higher than that of semantic analysis. In other words, the running efficiency of the mobile matrix method proposed in this paper surpasses that of the first two algorithms, enabling efficient generation of compound twills.

Furthermore, to compare the fitting accuracy of the three different algorithms, the error variations for each algorithm are individually calculated and presented in S5 Fig.

According to S5 Fig, the average fitting errors and standard deviation of three different algorithms can be obtained, as shown in S1 Table.

S5 Fig illustrates that the fitting error of both the Hilbert algorithm and the Bessel algorithm increases as the test step size grows. When the test step size reaches 8, the iterative process tends to stabilize. After testing, this iterative process is also applicable to other twill types. While initially higher than those of the other two algorithms, the fitting error of the moving matrix method proposed in this paper decreases consistently with further testing, resulting in an average error that is 33% lower than that of the other two algorithms. Consequently, it can be concluded that the proposed moving matrix method exhibits relatively high fitting accuracy.

To further verify the reliability of the algorithm proposed in this paper, representative fabric twills, namely double twill and triple twill, were selected for simulation experiments. The average fitting error and standard deviation of three different algorithms were calculated, and the results are presented in S2 Table and S3 Table.

As can be seen from S2 Table and S3 Table, the mobile matrix method proposed in this paper exhibits smaller simulation error indicators for other typical fabric twills compared to the other two algorithms, once again verifying that the fitting accuracy of the algorithm proposed in this paper is higher than that of traditional algorithms.

## Conclusions

This paper presents a novel compound twill generation algorithm based on a moving matrix, and the subsequent conclusions are derived through simulation and example verification.

1) When the number of twill flights changes, the matrix format of the initial twill also changes. Based on this, a moving matrix model is established, and the compound twill is formed through the iterative operation of the organizational cycle.

2) The primary algorithms are chosen for comparative experiments. The proposed algorithm demonstrates a 40% increase in efficiency compared to the Hilbert and Bessel algorithms, with an average error that is 33% lower than both, enabling efficient generation of compound twill. During the matrix operation process, it is also possible to improve cache hit rate and reduce memory access times by adjusting the order of loops, thereby further improving the operation efficiency.

3) The simulation effect of the method proposed in this article on other types of fabrics, such as jacquard fabrics, still has certain limitations. In the future, it can be considered to start from the geometric shape of the yarn, based on the brightness effect generated by lighting, establish a mathematical model for yarn simulation, and combine artificial intelligence technology to enhance the appearance of the fabric simulation as a whole. In addition, large-scale fabric simulation may require a large amount of memory resources, and a combination of sparse matrix and block computing can be used to effectively solve the memory and performance problems of moving matrix algorithms in large-scale fabric simulation.

## Supporting information

**S1-S5 Figs.  Figure-PLOS ONE - v3.**
(DOCX)

**S1 File.  Supporting Information - Lab Data - v2.**
(DOCX)

**S1-S3 Tables.  Table Information-PLOS ONE-v3.**
(DOCX)

## Acknowledgment

Xiaobo Yang expresses gratitude to the scientific research team at Zhejiang Shuren University.

## Author contributions

**Conceptualization:** Xiaobo Yang.

**Formal analysis:** Xiaobo Yang.

**Writing – original draft:** Xiaobo Yang.

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
