## [Decision Letter · Decision Letter 0]

15 Dec 2024

PONE-D-24-47803Research on Compound Twist Generation Algorithm Based on Moving MatrixPLOS ONE

Dear Dr. yang,

Thank you for submitting your manuscript to PLOS ONE. After careful consideration, we feel that it has merit but does not fully meet PLOS ONE’s publication criteria as it currently stands. Therefore, we invite you to submit a revised version of the manuscript that addresses the points raised during the review process.

Please note that we have only been able to secure a single reviewer to assess your manuscript. We are issuing a decision on your manuscript at this point to prevent further delays in the evaluation of your manuscript. Please be aware that the editor who handles your revised manuscript might find it necessary to invite additional reviewers to assess this work once the revised manuscript is submitted. However, we will aim to proceed on the basis of this single review if possible.  Could you please revise the manuscript to carefully address the concerns raised?

We look forward to receiving your revised manuscript.

Kind regards,

Helen Howard

Staff Editor

PLOS ONE

Journal Requirements:

The Zhejiang Province Natural Science Foundation, grant number Y1110023  

The Zhejiang Province Natural Science Foundation, grant number Y1110023, funded the study.

The Zhejiang Province Natural Science Foundation, grant number Y1110023

5. We note that your Data Availability Statement is currently as follows: All relevant data are within the manuscript and its Supporting Information files.

Reviewers' comments:

Reviewer's Responses to Questions

**Comments to the Author**

1. Is the manuscript technically sound, and do the data support the conclusions?

Reviewer #1: Partly

2. Has the statistical analysis been performed appropriately and rigorously? 

Reviewer #1: N/A

3. Have the authors made all data underlying the findings in their manuscript fully available?

Reviewer #1: Yes

4. Is the manuscript presented in an intelligible fashion and written in standard English?

Reviewer #1: No

5. Review Comments to the Author

Reviewer #1: This paper introduces a promising method for improving the simulation of complex fabrics, which has significant potential for textile engineering and CAD applications. With improvements in clarity, experimental detail, and discussion depth, this work could make a substantial contribution to the field.

1� The abstract could benefit from more clarity. For example, explain "changing flying number of twill curves" more explicitly to improve accessibility for readers less familiar with the terminology.

2� Include specific numerical results (e.g., fitting accuracy values) in the abstract to provide a concise summary of the contributions.

3� The introduction discusses previous work effectively but should better emphasize the limitations of current methods. This would help underline the novelty and significance of the proposed algorithm.

4� The mathematical derivations are comprehensive, but the explanation of key terms (e.g., "twill flight number") and their physical significance could be expanded to improve understanding.

5� Adding more visual aids (e.g., diagrams illustrating the moving matrix concept) would help clarify the methodology.

6� While the comparison with Hilbert and Bessel algorithms is compelling, more details on the experimental setup (e.g., test parameters and conditions) would enhance reproducibility.

7� Consider including a table summarizing the performance metrics for different algorithms to complement the figures.

8� The quality of the figures (e.g., Figure 2 and 3) is suboptimal. Higher-resolution images and better labeling would improve the reader's ability to interpret the results.

9� The discussion of results could be expanded to address potential limitations or constraints of the proposed algorithm (e.g., scalability for different fabric types).

10� Provide more insights into the practical implications of reduced error and increased efficiency for real-world applications.

11� The conclusion restates the findings well but could include a brief mention of future research directions, such as extending the algorithm to other textile structures or incorporating additional physical properties.

6. PLOS authors have the option to publish the peer review history of their article (what does this mean? ). If published, this will include your full peer review and any attached files.

**Do you want your identity to be public for this peer review?** For information about this choice, including consent withdrawal, please see our Privacy Policy .

Reviewer #1: No

---

## [Author Response · Author response to Decision Letter 1]

29 Dec 2024

I have corrected the manuscript as required and uploaded it to the system as an attachment

---

## [Decision Letter · Decision Letter 1]

11 Mar 2025

PONE-D-24-47803R1Research on Compound Twist Generation Algorithm Based on Moving MatrixPLOS ONE

Dear Dr. yang,

Thank you for submitting your manuscript to PLOS ONE. After careful consideration, we feel that it has merit but does not fully meet PLOS ONE’s publication criteria as it currently stands. Therefore, we invite you to submit a revised version of the manuscript that addresses the points raised during the review process. In particular, reviewers have raised concerns about the methodology and the discussion of the results.

We look forward to receiving your revised manuscript.

Kind regards,

Giulia Pascoletti, Ph.D.

Academic Editor

PLOS ONE

Reviewers' comments:

Reviewer's Responses to Questions

**Comments to the Author**

1. If the authors have adequately addressed your comments raised in a previous round of review and you feel that this manuscript is now acceptable for publication, you may indicate that here to bypass the “Comments to the Author” section, enter your conflict of interest statement in the “Confidential to Editor” section, and submit your "Accept" recommendation.

Reviewer #2: (No Response)

2. Is the manuscript technically sound, and do the data support the conclusions?

Reviewer #2: (No Response)

3. Has the statistical analysis been performed appropriately and rigorously? 

Reviewer #2: (No Response)

4. Have the authors made all data underlying the findings in their manuscript fully available?

Reviewer #2: (No Response)

5. Is the manuscript presented in an intelligible fashion and written in standard English?

Reviewer #2: (No Response)

6. Review Comments to the Author

Reviewer #2: 1- The paper compares the proposed moving matrix algorithm with Hilbert curve and Bézier curve generation algorithms, but no justification is provided as to why these algorithms were selected. Are they the most relevant benchmarks for evaluating twill generation? The selection process should be explained based on mathematical or computational similarities.

2- The experimental results are derived from a single set of fabric parameters. It is unclear whether the results generalize to other fabric types. Additional tests on various twill patterns and fabric densities are necessary to validate robustness.

3- The error reduction claim (i.e., 33% lower error compared to other methods) is based on an average error value. However, no standard deviation, confidence intervals, or statistical tests (such as ANOVA or t-tests) are provided to confirm the significance of these differences.

4- The iterative process of twill matrix transformations is not analyzed in terms of convergence. Does the matrix reach a steady-state representation over iterations, or does it exhibit chaotic behavior for certain twill types?

5- The paper repeatedly refers to the "twill flight number" but does not provide a clear definition or mathematical formulation. A more precise explanation, preferably with reference to textile engineering standards, is needed.

6- The study is entirely computational but does not attempt to validate the algorithm using real-world woven fabric structures. Were any physical textile samples tested against the predicted patterns to verify the accuracy of the generated twill structures?

7- The input structure for the moving matrix algorithm is not explicitly outlined. Does it require raw fabric images, predefined weave structures, or numerical parameters? A detailed data preprocessing section should be added.

8- The movement function defined in Equation (2) follows a simple translation transformation. However, why was this specific movement function chosen? Could alternative movement rules improve the algorithm’s flexibility?

9- The generated twill structures are presented as idealized models, but real-world fabric production involves factors like thread tension, material elasticity, and weave imperfections. How does the algorithm handle these practical constraints?

10- The method is optimized for compound twill weaves, but can it be extended to other complex patterns such as Jacquard, Satin, or Herringbone weaves? Some discussion on algorithm adaptability would be beneficial.

11- The paper does not mention any failure scenarios where the proposed method might not work effectively. Are there any boundary conditions where the algorithm’s performance deteriorates?

12- The proposed matrix-based approach might require significant memory resources for large-scale textile simulations. A discussion on memory efficiency and potential optimizations should be included.

7. PLOS authors have the option to publish the peer review history of their article (what does this mean? ). If published, this will include your full peer review and any attached files.

**Do you want your identity to be public for this peer review?** For information about this choice, including consent withdrawal, please see our Privacy Policy .

Reviewer #2: No

---

## [Author Response · Author response to Decision Letter 2]

20 Mar 2025

The document of response to reviewers has been uploaded.

---

## [Decision Letter · Decision Letter 2]

24 Mar 2025

Research on Compound Twist Generation Algorithm Based on Moving Matrix

PONE-D-24-47803R2

Dear Dr. yang,

We’re pleased to inform you that your manuscript has been judged scientifically suitable for publication and will be formally accepted for publication once it meets all outstanding technical requirements.

Kind regards,

Giulia Pascoletti, Ph.D.

Academic Editor

PLOS ONE

Additional Editor Comments (optional):

Reviewers' comments:

Reviewer's Responses to Questions

**Comments to the Author**

1. If the authors have adequately addressed your comments raised in a previous round of review and you feel that this manuscript is now acceptable for publication, you may indicate that here to bypass the “Comments to the Author” section, enter your conflict of interest statement in the “Confidential to Editor” section, and submit your "Accept" recommendation.

Reviewer #2: (No Response)

2. Is the manuscript technically sound, and do the data support the conclusions?

Reviewer #2: (No Response)

3. Has the statistical analysis been performed appropriately and rigorously? 

Reviewer #2: (No Response)

4. Have the authors made all data underlying the findings in their manuscript fully available?

Reviewer #2: (No Response)

5. Is the manuscript presented in an intelligible fashion and written in standard English?

Reviewer #2: (No Response)

6. Review Comments to the Author

Reviewer #2: (No Response)

7. PLOS authors have the option to publish the peer review history of their article (what does this mean? ). If published, this will include your full peer review and any attached files.

**Do you want your identity to be public for this peer review?** For information about this choice, including consent withdrawal, please see our Privacy Policy .

Reviewer #2: No

---

## [Editor Report · Acceptance letter]

PONE-D-24-47803R2

PLOS ONE

Dear Dr. Yang,

I'm pleased to inform you that your manuscript has been deemed suitable for publication in PLOS ONE. Congratulations! Your manuscript is now being handed over to our production team.

Kind regards,

on behalf of

Dr. Giulia Pascoletti

Academic Editor

PLOS ONE